# Open-Access 12-Minute MRI Screening for Acute Appendicitis: A Five-Year Retrospective Observational Study of Diagnostic Accuracy

**DOI:** 10.3390/jcm13237257

**Published:** 2024-11-28

**Authors:** Andrew Owen Jones, James Nol

**Affiliations:** 1Medical Imaging Department Blacktown Mount Druitt Hospitals, Blacktown Hospital, 18 Blacktown Road, Blacktown, Sydney, NSW 2148, Australia; 2School of Medicine, Western Sydney University, Sydney, NSW 2751, Australia; j.nol@westernsydney.edu.au

**Keywords:** appendicitis, appendicectomy, diagnostic imaging, magnetic resonance imaging, sensitivity, specificity, accuracy

## Abstract

**Objective**: This retrospective observational diagnostic accuracy study aims to demonstrate that open-access rapid-sequence non-contrast magnetic resonance imaging (MRI) is accurate for exclusion or confirmation of acute appendicitis (AA). **Methods**: In 2017, a locally designed 12 min MRI protocol was made available as a new open-access option (no booking needed) for any emergency department (ED) or acute surgical patient with any clinical presentation at the authors’ sites. Uninterrupted single-radiologist reporting availability was provided. A 5-year consecutive report list from 1 January 2019 to 31 December 2023 was recorded in an activity log, from which 3478 eligible reports were identified as ED-based referrals assessing for possible AA. There was “appendicitis possibility” in 581/3478 (17%) reports and “no evidence of appendicitis” in 2897/3478 (83%). These were retrospectively compared with the medical record findings of 557/3478 proven cases of AA (16%). Report availability and reliability metrics have been assessed. **Results**: Overall, 2583/3478 reports (74%) were finalized within 2 h of MRI study completion, 3254/3478 (94%) within 4 h. The 3478 reports combined had 98% sensitivity, 98% specificity, 98% accuracy, 94% positive predictive value, and 99% negative predictive value for AA (φ = 0.95). The largest 5-year subgroup, females 15–19 years old, 66/440 with proven AA, had 98% report accuracy. Pregnant women, 21/171 with proven AA, had 99% report accuracy. **Conclusions**: The described MRI protocol is accurate for appendicitis assessment and is a suitable first-imaging choice for children, young adults, and pregnant women. It does not require intravenous contrast and poses no radiation risk.

## 1. Introduction

Acute appendicitis (AA) is a common clinical concern, particularly during adolescence, when its incidence peaks. Retrospective analysis of USA data (1979–1984) revealed lifetime AA risks of 8.6% for males and 6.7% for females, with the highest occurrence observed between the ages of 10 and 19 years [1]. In this 1979–1984 analysis, prior to routine preoperative imaging, clinical accuracy of AA diagnosis, as measured by normal appendix removal rate at surgery (the “negative appendicectomy rate” [NAR]), was calculated to be 91.2% for males and 78.6% for females [1]. Pre-operative imaging aims to improve this accuracy to minimize NAR. Traditionally, clinicians have turned to either ultrasound (US) or computed tomography (CT); however, each has limitations.

A meta-analysis of 17 studies with 2841 participants found that US had a false negative rate of 55% and a false positive rate of 8% for detecting AA [2]. In a different study, the appendix could only be visualized using US in 89 (16%) of 562 cases. US accuracy for AA was 86.5% for these 89 “appendix visualized” cases alone; however, this reduced to only 13.7% when the other 473 “indeterminate” studies—where the appendix could not be visualized—were also considered [3]. Both publications recommended against routine use of US for AA assessment.

CT provides high diagnostic accuracy but has significant iatrogenic risk. A Cochrane review of 71 studies, involving a total of 10,280 participants, reported 95% sensitivity and 94% specificity for CT in diagnosing AA in individuals > 14 years old [4]. A 2010 publication showed beneficial NAR reduction from 23% to 1.7% when comparing data from the pre-routine CT era (1990–1994) to the period of 2003–2007 with >3000 pre-operative CT scans/year at the same institution [5]. BIER V (Biological Effects of Ionizing Radiation) data supported a commonly cited average population risk of 1 cancer death per 2000 CT studies [6]. Citing this BIER V 1:2000 risk assumption and reassessing the original published data from the abovementioned 2010 study, another author group predicted 1 CT-caused death for every 12 negative appendicectomies avoided in the 2010 study [7].

A 20-year-old female undergoing non-contrast CT of the abdomen and pelvis could face a cancer risk as high as 1 in 200, rather than 1 in 2000 [8]. If substituted into the above assumptions, this would predict nearly one CT-caused death for every negative appendicectomy avoided, with a 1:1 ratio reached at a CT risk of 1:167. The benefits of avoiding negative appendicectomy surgeries likely do not include lowered death risk, considering that a 10-year review of 16,315 laparoscopic emergency excisions of a normal appendix from United Kingdom national health records reported no deaths in patients < 49 years old [9].

In this study, we aimed to demonstrate that a locally designed 12 min non-contrast MRI abdomen protocol (“MRI screening”) has been accurate for diagnosing or excluding AA as the primary, and often only, imaging at the authors’ two emergency department (ED) sites, thereby avoiding both the radiation risk of CT and the poor accuracy risk of US. This has been available with extended hours every day since 2017 and has dramatically reduced both US and CT referrals for possible appendicitis assessment at each site.

## 2. Materials and Methods

### 2.1. Ethical Approval and Informed Consent

Ethical approval was obtained from the Western Sydney Local Health District Research & Education Network Scientific Advisory Committee (2311-07-QA), and the study was conducted in accordance with the principle outlined in the Declaration of Helsinki. As this was a retrospective observational quality assurance study, requirement for informed consent was waived.

### 2.2. Facility Overview

Blacktown Mount Druitt Hospital is a tertiary care hospital in New South Wales, Australia, operating across two campuses: one at Blacktown and one at Mount Druitt, Sydney. Each provides 24 h adult and pediatric emergency services. Within NSH Health, Blacktown is a “Major Hospital” transitioning to a “Principal Referral Hospital”, whereas Mount Druitt is a “District Hospital”. Clinical services are networked across the two sites with 530 beds combined. District records for 2023 showed that Blacktown Hospital had 65,000 presentations to ED, and Mount Druitt Hospital 40,000 presentations: 105,000 to ED across both sites, inclusive of all demographic groups. For both sites, Radiology Information System data for 2023 showed 90,275 referrals from ED to medical imaging, 15,528 for patients < 25 years old. Of these 15,528 referrals, 1963 were for MRI and 1309 were for CT. The 1963 MRI referrals included 1149 MRI screening abdomen referrals and 960 for possible appendicitis assessment. The 1309 CT referrals included 115 contrast CT abdomen pelvis for any reason, trauma included.

### 2.3. MRI Screening Abdomen Service Development

Since 2017, a locally designed MRI screening protocol has been available as an open-access imaging option (no booking or discussion needed) for any ED or acute surgical patient for any clinical presentation. ED staff simply completed an electronic request and safety sheet, notified MRI staff, and sent the patient for MRI. There was no directive to use this service; it just became a new imaging option. Any patient could be referred. Children < 5 years old and occasional patients > 80 years old were equally accepted without booking restriction for any clinical indication.

Adherence to uniform imaging protocol and report layout by a single reporting radiologist improved clinician familiarity. An activity log of all reports was maintained by A.O.J. on a single-user, encrypted, password-protected Microsoft Access database. This included the time and date of each report, whether AA was being considered, a summary of the report conclusion, and a field for later addition of a brief clinical outcome note. It was updated daily, including checks that no recent report had been missed. This activity log enabled follow-up of clinical outcomes, case comparison with similar cases, and development of a comprehensive appreciation of expected imaging appearances for all commonly encountered pathologies, including AA. It played a critical role in service development at our sites, independent of any planned review.

### 2.4. Data Collection

With date and data filters, a list of consecutive MRI screening reports fitting selection criteria could be produced from the activity log described above. These criteria were: report completed 1 January 2019 to 31 December 2023 inclusive; report finalized by A.O.J.; acute care ED-based referral; and AA a clinical consideration. The list was exported to a Microsoft Excel file for addition of further data fields for this review.

From an initial list of 4263 MRI screening reports within the defined timeframe, 3479 were identified when filtered to all requirements. One was excluded after review, as described below, leaving 3478 reports eligible for diagnostic accuracy analysis with no risk of significant missing data.

### 2.5. Use of Other Clinical Information

The reporting author had access to patient electronic medical records. The usual practice would be to quickly review ED presenting symptoms and available hematology results prior to reporting. Prior imaging was almost always not available or disregarded.

Description of presenting symptoms sometimes assisted with faster identification of appendix location or refined the range of considered alternative diagnoses. High inflammatory markers could prompt longer image review times. Neither had any further influence on the report conclusion.

Prior ultrasound imaging was considered unreliable and always ignored. CT and MRI, regarded by referring teams to have equivalent accuracy, were almost never performed together. For rare cases where an indeterminate CT had led to a subsequent MRI, the author would review MRI with little reference to the CT, as per the reason for that referral. Conversely, if CT were requested after MRI due to continued clinical uncertainty, the MRI had already been reported, and thus, its status in this review would not be affected (see Discussion).

Appendix report conclusion was always based on appendix MRI appearance alone. Achieving the best accuracy depended on this.

### 2.6. Equipment Siemens, Munich, Germany for All MRI Machines

Equipment at our campuses included Siemens 3T MRI units; Siemens Magnetom Trio Tim (from 2018) that was upgraded to Magnetom Prismafit (2022); Siemens Skyra 3T (from 2018); and Siemens Vida 3T (from 2020). All MRI machines were manufactured from Siemens, Munich, Germany.

### 2.7. Patient Preparation

Patients were instructed to drink two to three cups of water 20–45 min prior to imaging (longer is better). The MRI safety form was reviewed with the patient and the procedure explained.

### 2.8. Imaging Protocol (Four Sequences)

Coronal fast-acquisition (breath-hold) T2 image sets were obtained with and without fat suppression using TR 1500, TE 100, 3 mm thickness, no slice gap, and coverage from the pelvis to top of kidneys, angled along psoas muscles for best kidney coverage. Axial breath-hold T2 images were obtained through the lower abdomen and pelvis with TR 1500, TE 95, 5 mm thickness, and no slice gap. Sagittal breath-hold T2 images were obtained across the right side only with TR 1500, TE 100, 4 mm thickness, and no slice gap, extending from the pelvis to the top of the kidney (Figure 1). Total imaging time for all four sequences was 12 min. Fat-suppressed images can be identified by noting a dark subcutaneous fat appearance. Axial images are sometimes extended to the upper abdomen if symptoms extend to this level.

Optimal imaging was achieved with repeated small inspiratory breath holds. For young children or patients with reduced MRI tolerance, continued gentle breathing was allowed. This would cause mild image blurring but remained adequate for appendix assessment (Figure 1a).

### 2.9. Imaging Protocol Development

In May 2017, the authors decided to offset increasing requests for CT for appendix assessment in young patients by developing a new rapid MRI screening abdomen protocol to be offered as an open-access option for any emergency department (ED) or acute surgical patient for any indication. This began with a review of pre-existing 30 min general multipurpose MRI abdominal protocols, including one occasionally used for possible AA during pregnancy. T1, fat saturation T1, T2, and fat saturation T2 images with varying slice thickness and angulation were considered.

To be acceptable for pediatric patients, the imaging time needed to be <15 min. To be acceptable for diagnostic needs, the appendix, free fluid, free blood, female pelvic organs, and any area of inflammatory change had to be well demonstrated.

T1 imaging was dismissed. Significant hemorrhage was, in retrospect, always visible on T2 images. Axial, sagittal, and coronal breath-hold T2 images, with one additional series of fat saturation T2 images, satisfied requirements, provided that water oral contrast had been given to distinguish the small bowel. For time efficiency, slice-thicknesses were chosen such that the narrowest imaging plane (coronal) had the thinnest slice thickness (finest detail). Time used for sagittal images was halved by imaging only the symptomatic side. Fields-of-view were chosen so that upper abdomen (gallbladder, stomach, pancreas) to the lower pelvis (ovaries, free fluid) were covered in at least one plane. The coronal plane, having the highest resolution, was best for the additional fat saturation T2 series. Minor adjustments to slice thickness and angulation were made in 2017. By 2018, this protocol was established in its current form.

Diffusion-weighted imaging was tested for 90 cases after 2018, but was unhelpful and therefore was discontinued.

### 2.10. Recommended Image Interpretation Technique

The authors’ technique for image review was to start with the non-fat suppression sequences. Each image set was scrolled back and forth. On coronal images, the appendix was often identified crossing the iliac vessels. If not, the inferior cecum, toward the right kidney and near the right ovary, was checked. Caution was needed not to be misled by a Fallopian tube, round ligament, or lumbosacral plexus nerves. On axial images, a fleeting structure could often be identified between the cecum and psoas muscle. On sagittal images, the anterior psoas muscle margin, posterior margin of the ascending colon, and region anterior to the lumbosacral junction were checked. Fat suppression images generally only assisted with characterizing an inflamed appendix once found. If only one sequence was possible for a patient having difficulty, coronal T2 imaging without fat saturation was the best choice.

A request form suggestion of high or low suspicion of appendicitis was unhelpful. Specific details could be of assistance. Pain on urination could mean pyelonephritis, ureteric calculus, or an inflamed appendix against the bladder. Back pain could be due to spinal pathology or an inflamed appendix against the lower lumbar spine. Right iliac fossa pain was more likely due to an ovarian cyst if the appendix was shown on MRI to be high near the liver. Co-existing pathology was not uncommon. The appendix always needed to be directly assessed.

Diameter of the appendix was the starting point of assessment, but varied with luminal content and from base to tip. As with ultrasound and CT, where measuring the appendix diameter depended on appendix visibility, this was partially subjective. A measurement taken from the widest portion well clear of the base was generally best. A diameter of 10 mm near the base and 5 mm distally was usually normal, whereas a diameter of 5 mm near the base and 10 mm distally was usually AA (see Results).

Fluid adjacent to the appendix was common, and thus rarely helpful for AA assessment.

Appendix luminal fluid suggested a normal appendix if continuous with fluid in the cecum, but suggested AA if confined to only a distended distal appendix portion. Mild smooth fluid distension of the appendix lumen to the appendix base, occasionally seen due to cecum fecal loading, was regarded with caution, leading to further careful assessment for appendix wall thickening or mucosal contour irregularity.

Fecaliths within the appendix lumen were a very strong predicter of AA (see Results), particularly if distal appendix luminal fluid sharply defined the distal fecalith edge.

Appendix wall thickening and edema (brighter signal) strongly indicated AA; however, this did not always mean surgery was required. If continuous with similar changes along the ascending colon, appendicitis secondary to adjacent infective colitis, potentially responsive to a trial of medical therapy, was occasionally suggested.

Peri-appendiceal fat edema strongly implied AA only if surrounding it. Asymmetric adjacent edema could sometimes be due to nearby cecal diverticulitis or epiploic appendagitis.

To achieve high report accuracy, all factors above needed to be considered. Appendix diameter was only the starting point.

### 2.11. Statistical Analysis

#### 2.11.1. MRI Report Classification

Each MRI report was categorized as either “any possibility positive” (positive; 581/3478 reports) or “no evidence of appendicitis” (negative; 2897/3478 reports). Suggested alternative diagnoses were disregarded. There were no “indeterminate” reports. According to the standardized approach, any case with uncertainty had the conclusion “Appendicitis could still be considered”, which was regarded as a positive report for AA by clinicians and a positive report for AA for this review (see Discussion).

#### 2.11.2. Clinical Outcome Classification

Clinical outcomes were assessed by reviewing medical records at least 2 weeks after patient discharge. A patient presentation was regarded to have been clinically negative for AA (2921 presentations) in the presence of clinical improvement without surgery or intravenous antibiotics and no clinical relapse (2811/2921), laparoscopy performed for other reasons, appendix left in place (58/2921), or appendicectomy performed but no AA on pathology report (52/2921).

A patient presentation was regarded to have been clinically positive for AA (557 presentations) if appendicectomy had been performed and AA was confirmed on the pathology report (536/557), if an abscess at the appendix site had been successfully treated (6/557), if the operative report description confirmed AA (4/557), if there had been strong clinical agreement and successful medical treatment with intravenous antibiotics (6/557), if there had been florid positive MRI imaging changes with clinical agreement but went elsewhere for surgery (4/557), or if an initial missed diagnosis had been proven at a second presentation (1/557).

#### 2.11.3. Statistical Classification

A true positive report is where appendicitis was suggested, and the clinical outcome was AA. A false positive report is where appendicitis was suggested, and the clinical outcome did not include AA. A true negative report is where a report directly stated, “there is no evidence of appendicitis” (or similar), and the clinical outcome did not include AA. A false negative report is where a report directly stated, “there is no evidence of appendicitis” (or similar), and the clinical outcome was AA. One patient, who returned 3 days after discharge with ongoing symptoms, had undergone two MRI studies, and hence had two report entries: (i) false negative; and (ii) true positive. All other entries were for unrelated presentations.

Each image series was assessed by the same radiologist (A.O.J.) using consistent report phrasing. Each report concluded with a clear statement of either “possibility of appendicitis” (level of certainty specified) or “no evidence of appendicitis” (or similar). For a few difficult or uncertain cases, the phrase “appendicitis could still be considered” was used, recorded as a positive report in this series. There were no “indeterminate” reports.

One presentation within the initial activity log list of 3479 reports could not be appropriately classified. A 26-year-old female with right flank pain for 3 days had had a dilated fluid-signal appendix on MRI, reported as possible appendicitis. Pathology described a low-grade appendix mucinous neoplasm with clear margins. The “false positive” criteria were not met, as appendicectomy was curative. However, the “true positive” criteria were also not met, as there was no acute inflammation. This presentation was excluded from this series, leaving 3478 presentations for accuracy performance assessment.

#### 2.11.4. Statistical Performance Measures

Accuracy:$$\left(\frac{\left(TP+TN\right)}{\left(TP+FP+FN+TN\right)}\right)\times 100\%$$

Rounded down to prior centile integer (disadvantageous).

Best value 100%.

F_1_ Score:$$\left(\frac{\left(2\times TP\right)}{\left(\left(2\times TP\right)+FP+FN\right)}\right)$$

Rounded down to the prior hundredth value (disadvantageous).

Best value = 1.

Negative Likelihood Ratio:$$\frac{\left(1-\left(\frac{TP}{TP+FN}\right)\right)}{\left(\frac{TN}{FP+TN}\right)}$$

Rounded up to the next hundredth value (disadvantageous).

Best value = 0.

Negative Predictive Value:$$\left(\frac{TN}{FN+TN}\right)\times 100\%$$

Rounded down to the prior centile integer (disadvantageous).

Best value = 100%.

Phi Coefficient (φ):$$\left(\frac{\left(\left(TP\times TN\right)-\left(FP\times FN\right)\right)}{\sqrt{\left(TP+FP\right)(TP+FN)(FP+TN)(FN+TN)}}\right)$$

Rounded down to the prior hundredth value (disadvantageous).

Best value = 1.

Positive Likelihood Ratio:$$\frac{\left(\frac{TP}{TP+FN}\right)}{\left(1-\left(\frac{TN}{FP+TN}\right)\right)}$$

Rounded down to the prior integer value (disadvantageous).

False positive (FP) 0 causes division by zero error (“/0!”).

Positive likelihood ratio > 10 regarded suitable for a screening test.

Positive Predictive Value (Precision):$$\left(\frac{TP}{TP+FP}\right)\times 100\%$$

Rounded down to the prior centile integer (disadvantageous).

Best value = 100%.

Prevalence:$$\left(\frac{\left(TP+FN\right)}{\left(TP+FP+FN+TN\right)}\right)\times 100\%$$

Rounded down to the prior centile integer (disadvantageous).

Prevalence 0% would bias true negative rate to 100%.

Prevalence 100% would bias true positive rate to 100%.

Prevalence 50% would not favor either measure.

Prevalence Threshold (see Appendix A):$$\left(\frac{\sqrt{\left(\frac{TP}{TP+FN}\right)\times \left(\frac{FP}{FP+TN}\right)}-\left(\frac{FP}{FP+TN}\right)}{\left(\frac{TP}{TP+FN}\right)-\left(\frac{FP}{FP+TN}\right)}\right)\times 100\%$$

Rounded up to the next centile integer (disadvantageous).

Prevalence threshold < prevalence reduces false positives.

True Negative Rate (Specificity):$$\left(\frac{TN}{\left(FP+TN\right)}\right)\times 100\%$$

Rounded down to the prior centile integer (disadvantageous).

Best value = 100%.

True Positive Rate (Sensitivity):$$\left(\frac{TP}{\left(TP+FN\right)}\right)\times 100\%$$

Rounded down to the prior centile integer (disadvantageous).

Best value = 100%.

## 3. Results

### 3.1. Statistical Performance Outcomes

Of all reports, 2889/3478 were true negative, 549/3478 were true positive, 8/3478 were false negative, and 32/3478 were false positive for AA. With disadvantageous rounding to prior centile, these reports had 98% sensitivity, 98% specificity, 98% accuracy, 94% positive predictive value, and 99% negative predictive value (φ = 0.95) for AA (Table 1).

Within the female cohort, 171/2295 (7%) were referred for possible appendicitis assessment during pregnancy. These reports had 100% sensitivity and 99% specificity for detecting AA (Table 2).

### 3.2. Service Hours

Between January 2019 and February 2023, MRI service hours were from 07:30 AM to midnight every day, with occasional cases after midnight due to staff staying back. During March 2023, the service hours were extended to 24 h coverage every day (Figure 2). Of all scans, 840/3478 (24%) were performed on Saturdays or Sundays.

### 3.3. Urgent Report Availability

MRI screening was typically the only imaging performed for patients presenting to ED with possible AA; thus, reports needed to be finalized within an acceptable timeframe. For this series, 2583/3478 reports (74%) were finalized within 2 h of MRI completion, 3254/3478 (94%) within 4 h (Figure 3). Due to the availability of only a single radiologist, when overnight service commenced from March 2023, clinically stable patients scanned overnight would have their reports delayed until the next morning, typically before 8:00 a.m., unless an immediate report was requested. This was acceptable for our ED services.

### 3.4. Increasing Demand for Abdominal MRI Screening Assessment

Figure 2 demonstrates progressively increasing uptake of MRI screening as an imaging option, including increasing uptake of overnight imaging requests with the commencement of 24 h availability from March 2023. Figure 3 shows that, despite more numerous report delays > 4 h for overnight studies, this did not impede continued increasing referral rates at the authors’ sites.

### 3.5. NAR of the Study

Of the 588 patients with anatomical pathology in this series, 52/588 were negative for AA, and 536/588 were diagnosed with AA. The NAR for this series was, thus, 8.8% (52/588).

### 3.6. Appendix Diameter

In this series, a reported appendix diameter ≤ 4 mm had 100% negative predictive value for AA, while a diameter ≥ 11 mm had 100% positive predictive value. A 7 mm measurement was less helpful, having an AA outcome in 82/179 reports (46%) and a normal outcome in 97/179 reports (54%) (Table 3). For 13/3478 (<1%) presentations, the appendix could not be measured due to appendix replacement by abscess, other pathology such as intussusception, or simple inability to visualize.

### 3.7. Fecaliths

A fecalith was reported in 154/3478 (4%) of MRI reports. Of these, 152/154 (99%) had true positive reports. In one false positive report, a fecalith had been reported as though within an appendix, in retrospect likely within an adjacent inflamed cecal diverticulum. In one true negative report, a small non-obstructing fecalith was reported as an incidental finding within an otherwise normal appendix. Two of eight false negative reports in this series also, in retrospect, had a fecalith within a dilated appendix. In one case, the dilated appendix and fecalith were, in retrospect, clearly visible in a different location to that originally described (reader perception error). In the other case, the appendix and fecalith had each been seen, but had been mistaken to be the small bowel containing a gas bubble (reader interpretive error).

### 3.8. Incidental Neuroendocrine Tumors with Co-Existent AA

Pathology identified four incidental fully resected appendix neuroendocrine tumors, measuring 12 mm, 7 mm, 2 mm, and 1 mm, all with co-existing AA. None of these tumors were identified prospectively on MRI amongst the background inflammatory change.

### 3.9. Co-Existent Enterobius Vermicularis (Pinworm) Infestation

Pinworm infestation was identified on histology of a removed appendix for seven patients. Three (10, 12, and 14 years old) had MRI reports suggesting appendicitis and confirmed AA on histology in addition to pinworm infestation (true positive report). One (6 years old) had an MRI report suggesting early appendicitis (7 mm distal diameter, luminal fluid containing debris), histology showing only pinworm, and no acute inflammation (false positive report). Three (ages 13, 20, and 21 years) had MRI reports of “no evidence of appendicitis” with histology showing only pinworm infestation, with no acute inflammation (true negative reports). The 21-year-old had CT after MRI, which was also reported as normal. All patients had uneventful recoveries following surgery. None had pinworm either clinically suspected or prospectively identified on MRI imaging.

### 3.10. Chronic Inflammatory Change on Histology Review

A histology review of 588 appendicectomy specimens showed acute inflammatory infiltrate (AA) in 536/588. Of these, 479/536 (89%) had only acute inflammatory infiltrate, 33/536 (6%) had acute-on-chronic inflammatory components, 14/536 (3%) had background old fibrous obliteration of the tip, 4/536 (<1%) had incidental neuroendocrine tumor, 3/536 (<1%) had pinworm infestation, and 1/536 (<1%) showed mucosal metaplasia suggestive of possible early mucocele in additional to florid acute inflammation. Of the 52 appendicectomy specimens negative for AA, 39/52 (75%) showed normal histology, 9/52 (17%) showed background old fibrous obliteration of the tip or lumen, and 4/52 (8%) showed only incidental pinworm infestation.

### 3.11. Retrospective Review of False Negative Errors

Of eight false negative reports, 3/8 were reader perception errors, where, in retrospect, an inflamed appendix portion could be readily seen in another location to that reported; 1/8 were the interpretative error described above; and for 4/8, AA remained difficult to appreciate, even in retrospect (see Discussion). All initial errors are included in this review.

### 3.12. Female to Male Comparison

Females were more frequently referred for MRI in this series (2295/3478, [66%]) than males (1183/3478, [34%]) due to the broader range of female-specific alternative diagnoses that could clinically mimic AA. This resulted in lower AA prevalence in female referral subgroups compared to males. The highest female subgroup prevalence of AA by 5-year age group was 28/148 (19%) for ages 35–39 years, then 66/440 (15%) for 15–19 years. This compared to a male subgroup prevalence of AA 72/285 (25%) in the age range of 9–14 years and >25% for all older 5-year age groups up to 40 years (Table 1).

### 3.13. Imaging Young Children Without Sedation

Almost all young children completed this MRI protocol without sedation. Neither intravenous sedation nor general anesthesia was offered. Of the 65 children < 4 years old, 3 had supervised oral chloral hydrate sedation, while 62 successfully completed the procedure without it. Among 78 children 5 years old, 3 had supervised oral chloral hydrate sedation, and 75 did not. Children > 5 years old were routinely imaged without sedation. If timed to a normal sleep pattern, babies could be imaged after feed and wrapping during a normal sleep. For young children, some examinations could be shortened by omitting one or two sequences without losing diagnostic value. For many, allowing continued quiet breathing during imaging, rather than use of breath-holds, significantly improved tolerance with minimal image blurring. All MRI studies, complete or not, were included in this series, provided that at least one sequence could be obtained. Coronal T2 imaging was prioritized if tolerance appeared to be reduced.

## 4. Discussion

MRI screening for AA had 98% accuracy for 3478 reports. There were no known adverse effects from MRI for any patient. No intravenous contrast or line disposables were cost savings compared to CT. The examination time was 12 min compared to 20 min for US. Time-to-report and reporting costs had few differences between modalities.

This protocol had ≥98% negative predictive value for AA for all age groups. A high negative predictive value, however, typically increases clinicians’ reliance on this result, with increased risk of missed AA diagnosis for the rare unexpected remaining false negative reports. To illustrate this, the eight false negative reports from this series are described.

(1)In May 2019, a 4-year-old male was referred with fever, copious watery diarrhea, reduced oral intake, and vomiting. An initial ultrasound was inconclusive, recommending MRI. MRI reported severe intra-abdominal inflammatory change, appendix not visualized, likely severe enterocolitis. The patient was admitted on intravenous antibiotics. Day-2 repeat (ward-referred) MRI screening examination suggested severe appendicitis. Day-3 laparoscopy confirmed perforated AA. On retrospective review of the initial MRI, all changes described in the operation report were visible, including the perforated inflamed appendix at a different location to that described (reader perception error).(2)In October 2020, a 37-year-old female was referred with nausea and right iliac fossa pain. Ultrasound (appendix not seen) was followed by MRI (reported normal). Due to ongoing symptoms, same-day CT was also performed. CT suggested early AA, confirmed at surgery. In retrospect, subtle changes of tip appendicitis were evident on the MRI.(3)In July 2021, a 24-year-old male was referred with migratory right iliac fossa pain, nausea, and decreased appetite. MRI was the first investigation, and was reported normal. CT was performed next morning, showing likely early AA, confirmed at surgery. MRI diagnosis remained difficult, even in retrospect.(4)In July 2021, a 19-year-old female was referred with peritonitic abdomen. MRI was the first investigation, and was reported normal. Due to ongoing symptoms, same-day CT was performed, reporting early AA, confirmed at surgery. In retrospect, an inflamed appendix was readily evident at the left (rather than right) iliac fossa, as shown in Figure 1b (reader perception error).(5)In December 2021, a 17-year-old female was referred with right iliac fossa pain, diarrhea, and raised white cell count. MRI was the first investigation, and was reported normal. Ultrasound the next day was reported normal (appendix not seen). Due to ongoing symptoms, laparoscopic appendicectomy was performed that day. The appendix had a normal operative appearance. Despite this, histology showed AA, with patchy transmural acute inflammation. MRI diagnosis remained difficult, even in retrospect.(6)In February 2022, a 21-year-old female was referred with right iliac fossa pain. MRI was the first investigation, showing a 6 cm abscess at the upper aspect of the right ovary, mistakenly reported as a tubo-ovarian abscess. An interventional radiologist reviewed the images and suggested that an inflamed appendix could be seen extending to this site. The author agreed and added a corrective addendum (reader perception error). Treatment was not interrupted.(7)In January 2023, a 22-year-old female was referred with severe right colicky abdominal pain and right-sided tenderness. MRI was the first investigation, reporting gallstones with no bile duct dilatation or pericholecystic inflammation, appendix normal. Ultrasound next day reported probe tenderness over the gallbladder, appendix not assessed. Next-day laparoscopy showed gallbladder distension and AA. Pathology showed chronic cholecystitis and AA. In retrospect, MRI showed minimal edema along a normal diameter appendix.(8)In August 2023, a 15-year-old male was referred with recent-onset vomiting and loose stools, rigid but non-tender abdomen. MRI was the only imaging, reported as mesenteric adenitis. He was discharged home. Three days later, he represented to ED with fever and worsened abdominal pain. A second ED-referred MRI showed severe appendicitis with tip perforation, confirmed at laparoscopic appendicectomy. In retrospect, the first MRI showed dilated tip appendicitis with a contained fecalith, mistaken to be small bowel containing a gas bubble (reader interpretive error).

These cases demonstrate the importance of clearly identifying the appendix in at least two imaging planes for every report. It is recommended that image numbers showing the appendix be included within each report. Whenever the appendix has not been clearly identified, “appendicitis remains a consideration” should always be the conclusion. False negative reports carry the highest risk of harm and must be avoided whenever there is doubt.

Many metrics of diagnostic accuracy performance have been presented. Metrics of prevalence threshold, F_1_ score, and Phi coefficient will be briefly discussed.

Prevalence threshold calculates the AA prevalence below which the positive predictive value for MRI would be expected to sharply decline [10]. Patients 5–9 years old were unfavorable on this measure, with 6% AA prevalence, significantly lower than the 10% calculated prevalence threshold. This group had 85% positive predictive value. Older age groups had favorable threshold values, each below the corresponding AA prevalence. These groups each had over 90% positive predictive value (Table 1). The prevalence threshold does not, however, relate directly to the primary purpose for MRI screening, which was to exclude, rather than confirm, AA. MRI screening had 100% negative predictive value for the 5–9-years age group, fully satisfying this referral goal. Furthermore, for all 3478 reports, the threshold value of 10% for all reports combined was favorable compared to the overall 16% AA prevalence, with 94% combined positive predictive value, favorable as a screening test metric.

F_1_ score and Phi coefficient are metrics intended to compare different studies with different disease prevalence rates. An important difference is that F_1_ score does not include true negative results, whereas Phi coefficient includes all outcomes and could, thus, be considered the more wholistic metric [11]. To illustrate this, we present 3478 reports with an F_1_ score of 0.96 for AA identification (“true positive” = correct AA report). The same reports would have an F_1_ score of 0.99 for normal appendix identification (“true positive” = correct normal appendix report). Phi coefficient is unchanged at 0.95 regardless of true positive definition. Both are provided for reference; however, Phi coefficient is preferred by the authors over F_1_ score as the better performance metric.

To achieve referrer confidence in MRI screening, a consistent imaging protocol, consistent report presentation, prompt reporting, open communication, reliable critical results notification procedures, and providing the best possible outcomes were essential. One study showed US sensitivities for AA in the first, second, and third trimesters of pregnancy of 69%, 63%, and 51%, respectively, with corresponding specificities of 85%, 85%, and 65% [12]. MRI screening for 171 pregnant patients had 100%, 100%, and 100% sensitivity, and 98%, 98% and 100% specificity for the first, second, and third trimesters of pregnancy, in addition to providing 100% negative predictive value all 1231 children in both the 5–9-year and 9–14-year-old subgroups combined.

A 2020 publication suggested that routine pre-operative CT could reduce NAR from 22% to 7%, with an 89% reduction in healthcare costs and better allocation of health resources elsewhere [13]. This claim disregards potential for delayed costs due to radiation-induced harm and incorrectly presents NAR as a CT diagnostic accuracy metric for exclusion of AA, rather than sensitivity or negative predictive value.

A South Korean review of 825,820 preoperative appendix CT assessment studies (52.9% male; median age, 28 years) showed a 1.26 times higher risk of developing delayed hematologic malignant neoplasms (mostly myeloid leukemia), including a 2.14 times higher risk for those aged 0–15 years [14]. A European study showed an increased risk of hematologic malignancy within 12 years after any CT for patients < 22 years old (1–2 persons per 10,000) [15]. An Australian study of 10.9 million people aged 0–19 years found a 24% higher cancer risk following just one CT [16]. An American study calculated the mean CT-associated cancer risk for a single non-contrast CT of the abdomen and pelvis to be 1 in 500 (for females) or 1 in 660 (for males) at age 20 years [8]. CT intravenous contrast has additional risks and costs [17]. A high likelihood for delayed healthcare costs would be difficult to quantify in cost, but should not be disregarded.

NAR is not a diagnostic accuracy study metric. It is the proportion of patients who had a normal appendix removed at surgery. Local policy recommendations, individual clinician experience, patient demographic considerations, clinician confidence in imaging reports, and patient preference all can influence this. Imaging sensitivity is an appropriate metric without these confounding factors. On meta-analysis review, CT had 95% sensitivity for AA in patients > 14 years age [4]. MRI screening in this review had 98% sensitivity, including pediatric and pregnant patients. These sensitivity results could be considered equivalent. NAR should not be used to make comparison across sites, between modalities, or for different demographic groups. It includes too many confounding variables. Our data showed 8.8% NAR for MRI screening. This, however, was shared equally across all patient groups typically excluded from CT.

In the 3478 cases assessed, confident alternative diagnoses provided included cecal diverticulitis, epiploic appendagitis, enterocolitis, mesenteric adenitis, dermoid cyst, hemorrhagic ovarian cyst, peptic ulceration, severe corpus luteum hemorrhage, tubal or ovarian torsion, pyosalpinx, ectopic pregnancy, pyelonephritis, ureteric calculus, cholecystitis, bile duct calculus, pancreatitis, and intussusception. Neither CT nor US alone have this diagnostic range.

### Limitations

A study design limitation is that assessments for all MRI reports were conducted by a single radiologist (A.O.J.). Other studies would be needed to confirm similar results with other reporters. To the best of our knowledge, there is no comparable single-site large series review.

A 2021 Cochrane review of MRI in pregnant women, children, and adults showed 95% summary sensitivity and 96% summary specificity for AA diagnosis [18]. This included 1980/7462 participants with AA from 58 studies but highlighted frequent methodology limitations of these studies. Our report added a large series to the literature, but did not avoid all methodological concerns raised. Of these, the most significant limitation of our review was reliance on case note review for all non-surgical clinical outcome conclusions, as was commonly encountered in the Cochrane review. This has significant potential to cause positive study bias. Substituting worst-case assumptions illustrates this.

For the outcome “appendicitis”, our study had 557/3478 presentations. Of these, 11/557 were based on assumptions: one false negative based on a later presentation; six true positive based on response to intravenous antibiotics rather than surgery; and four true positive based on “convincing” MRI changes, initial clinical agreement and going elsewhere for surgery with no further verification. To have excluded these would have disrupted the consecutive design of this review. For the worst-case scenario that all 11/11 should be false positive, the adjusted totals of 539 true positive versus 42 false positive reports would still result in 98% sensitivity, 98% specificity, and 98% accuracy for our cohort, protected by the large sample size.

For the outcome “not appendicitis”, our study had 2921/3478 presentations, 2811 of which were true negative on the assumption that none had represented with AA diagnosis to another site using a different medical records system. For the worst-case scenario that all had presented to another site with missed AA, with no feedback returned for this ongoing issue, the adjusted total of 2819 false negative reports versus 78 surgically confirmed true negative reports would result in 16% sensitivity, 70% specificity, and 18% accuracy for our cohort, not protected by the large sample size. Although it is an unrealistic scenario, this highlights the Cochrane review criticism that reliance on reviewing notes for non-surgical follow-up is a substantial methodology weakness. Unfortunately, the much better alternative of follow-up telephone contact with every patient was beyond the resources available to the authors, and hence could not be within the study design scope.

A subgroup statistical limitation of this study is that the sample size for the youngest age group, 0–4 years old, was too small to be statistically relevant. Although these results can be incorporated into total cohort analysis without issue, meaningful subgroup analysis is not possible for this 0–4-year-old age range individually, as evidenced by the reduced Phi coefficient for the subgroup total and by the unrealistic discrepancy in all metrics when comparing males and females within this subgroup. Further studies would be needed to overcome this.

As a final limitation, we describe in detail a specific protocol using specific equipment. Further studies would be worthwhile to show an ability to effectively transfer this to other sites with other equipment. Mild adjustments to some parameters would likely be needed at some sites.

## 5. Conclusions

The described 12 min rapid-sequence, non-contrast MRI protocol has been shown to be accurate for AA assessment. This large series review validates this MRI screening approach as a viable alternative to CT with none of the associated contrast or radiation risks. MRI screening at the authors’ sites has already achieved diagnostic outcomes for AA equivalent to CT, within the same clinical timeframe, without any ionizing radiation or contrast risk.

## Figures and Tables

**Figure 1 jcm-13-07257-f001:**
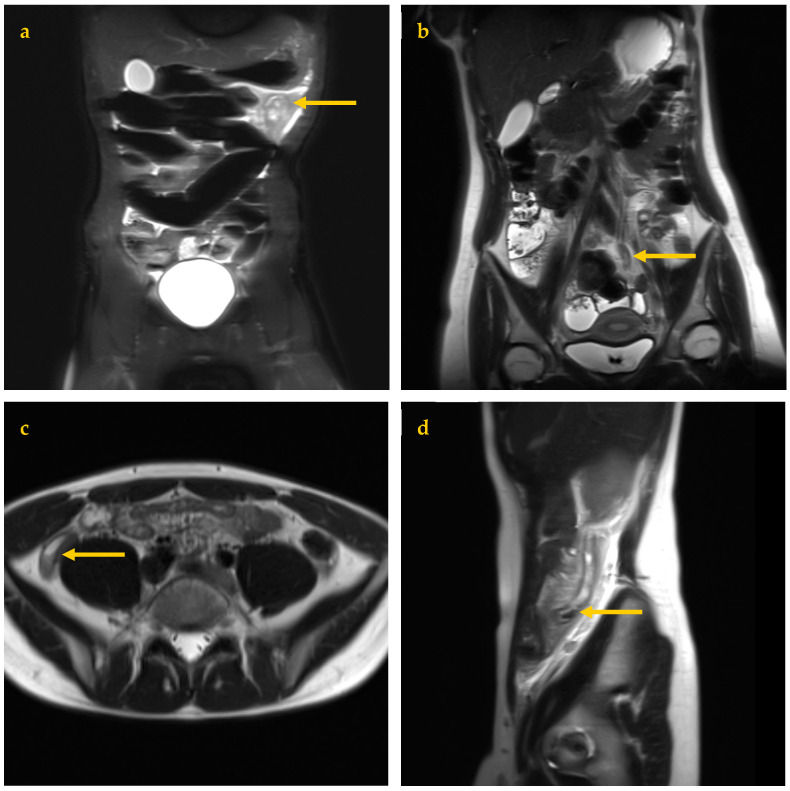
The four MRI sequences used, each demonstrating a different example of proven AA (arrows). (**a**) Coronal Fat Saturation T2 image, mildly degraded by breathing movement in a 12-year-old male showing bright edema signal due to proven left upper quadrant AA. (**b**) Coronal T2 image of a 19-year-old female showing a mildly thickened distal appendix extending to the left lower quadrant. (**c**) Axial T2 image of an 18-year-old male showing typical right lower quadrant AA, including fluid and debris layering within the appendix lumen. (**d**) Sagittal T2 image of a 14-year-old female showing an inflamed appendix along the right flank with a luminal appendicolith (arrow).

**Figure 2 jcm-13-07257-f002:**
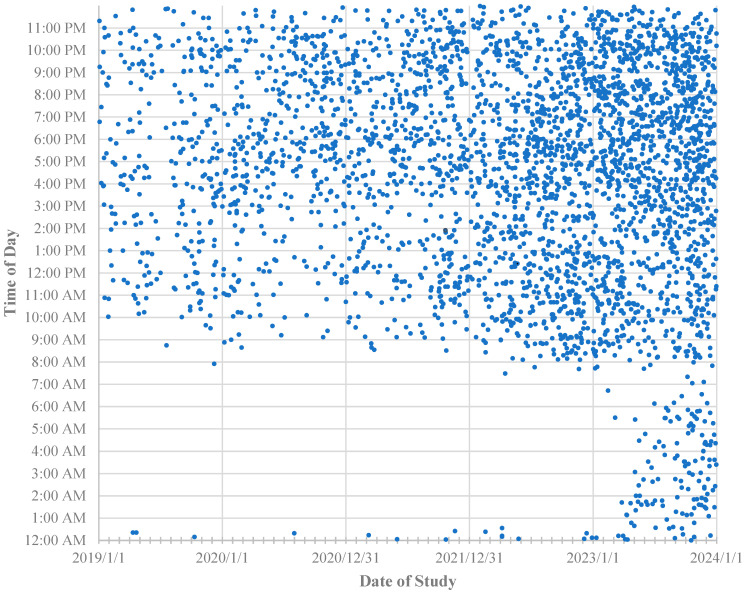
Abdominal MRI screening: open-access availability for ED. Times of service for all 3478 studies between 1 January 2019 and 31 December 2023 are displayed. Service hours were expanded from 07:30 AM to midnight 7 days per week to 24 h availability 7 days per week in March 2023. Increasing demand is demonstrated by progressively increasing point density.

**Figure 3 jcm-13-07257-f003:**
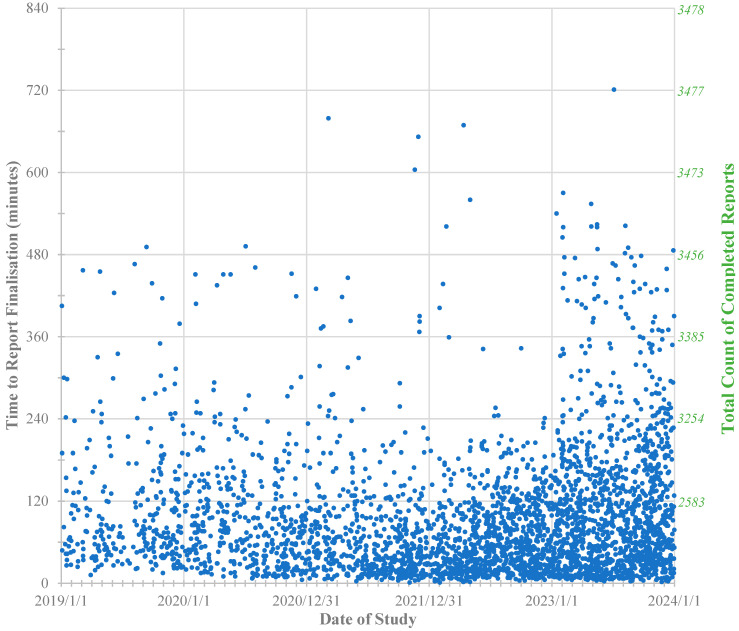
Abdominal MRI screening: Time from MRI study completion to report finalization in minutes for all 3478 studies conducted between 1 January 2019 and 31 December 2023. The longest delays were due to errors in radiologist notification for clinically stable patients with additional delayed clinical escalation request.

**Table 1 jcm-13-07257-t001:** Analysis of all reports: report prediction versus clinical conclusion.

Age (Years)	M	F	All	M	F	All	M	F	All	M	F	All	M	F	All
	**Total** **Reports**			**Outcome** **AA**			**Outcome ** **Not AA**			**Prevalence of** **AA**			**Prevalence** **Threshold**		
0–4	32	33	65	1	3	4	31	30	61	3%	9%	6%	100%	0%	13%
5–9	333	289	622	30	11	41	303	278	581	9%	3%	6%	11%	10%	10%
10–14	285	324	609	72	42	114	213	282	495	25%	13%	18%	13%	11%	12%
15–19	195	440	635	59	66	125	136	374	510	30%	15%	19%	11%	11%	11%
20–24	126	412	538	38	47	85	88	365	453	29%	11%	15%	0%	8%	7%
25–29	105	322	427	38	37	75	67	285	352	36%	11%	17%	11%	6%	8%
30–34	61	223	284	26	32	58	35	191	226	42%	14%	20%	0%	12%	11%
35–39	28	148	176	9	28	37	19	120	139	32%	19%	21%	0%	12%	11%
≥40	18	104	122	3	15	18	15	89	104	16%	14%	14%	0%	0%	0%
**All**	**1183**	**2295**	**3478**	**276**	**281**	**557**	**907**	**2014**	**2921**	**23%**	**12%**	**16%**	**11%**	**10%**	**10%**
	**Report ** **Suggested** **Appendicitis**			**True Positive (TP)**			**False Positive (FP)**			**Positive** **Predictive Value (Precision)**			**Negative** **Predictive Value**		
0–4	1	3	4	0	3	3	1	0	1	0%	100%	75%	96%	100%	98%
5–9	34	14	48	30	11	41	4	3	7	88%	78%	85%	100%	100%	100%
10–14	76	46	122	72	42	114	4	4	8	94%	91%	93%	100%	100%	100%
15–19	60	69	129	58	64	122	2	5	7	96%	92%	94%	99%	99%	99%
20–24	37	47	84	37	45	82	0	2	2	100%	95%	97%	98%	99%	99%
25–29	39	38	77	38	37	75	1	1	2	97%	97%	97%	100%	100%	100%
30–34	26	35	61	26	32	58	0	3	3	100%	91%	95%	100%	100%	100%
35–39	9	29	38	9	27	36	0	2	2	100%	93%	94%	100%	99%	99%
≥40	3	15	18	3	15	18	0	0	0	100%	100%	100%	100%	100%	100%
**All**	**285**	**296**	**581**	**273**	**276**	**549**	**12**	**20**	**32**	**95%**	**93%**	**94%**	**99%**	**99%**	**99%**
	**Reported** **Not** **Appendicitis**			**False Negative (FN)**			**True Negative (TN)**			**Positive** **Likelihood ** **Ratio**			**Negative** **Likelihood ** **Ratio**		
0–4	31	30	61	1	0	1	30	30	60	0	/0!	45	1.04	0	0.26
5–9	299	275	574	0	0	0	299	275	574	75	92	83	0	0	0
10–14	209	278	487	0	0	0	209	278	487	53	70	61	0	0	0
15–19	135	371	506	1	2	3	134	369	503	66	72	71	0.02	0.04	0.03
20–24	89	365	454	1	2	3	88	363	451	/0!	174	218	0.03	0.05	0.04
25–29	66	284	350	0	0	0	66	284	350	67	285	176	0	0	0
30–34	35	188	223	0	0	0	35	188	223	/0!	63	75	0	0	0
35–39	19	119	138	0	1	1	19	118	137	/0!	57	67	0	0.04	0.03
≥40	15	89	104	0	0	0	15	89	104	/0!	/0!	/0!	0	0	0
**All**	**898**	**1999**	**2897**	**3**	**5**	**8**	**895**	**1994**	**2889**	**74**	**98**	**89**	**0.02**	**0.02**	**0.02**
	**True Positive Rate** **(Sensitivity)**			**True Negative Rate** **(Specificity)**			**Accuracy**			**F_1_ Score**			**Phi Coefficient (φ)**		
0–4	0%	100%	75%	96%	100%	98%	93%	100%	96%	0	1	0.75	−0.04	1	0.73
5–9	100%	100%	100%	98%	98%	98%	98%	98%	98%	0.93	0.88	0.92	0.93	0.88	0.91
10–14	100%	100%	100%	98%	98%	98%	98%	98%	98%	0.97	0.95	0.96	0.96	0.94	0.95
15–19	98%	96%	97%	98%	98%	98%	98%	98%	98%	0.97	0.94	0.96	0.96	0.93	0.95
20–24	97%	95%	96%	100%	99%	99%	99%	99%	99%	0.98	0.95	0.97	0.98	0.95	0.96
25–29	100%	100%	100%	98%	99%	99%	99%	99%	99%	0.98	0.98	0.98	0.97	0.98	0.98
30–34	100%	100%	100%	100%	98%	98%	100%	98%	98%	1	0.95	0.97	1	0.94	0.96
35–39	100%	96%	97%	100%	98%	98%	100%	97%	98%	1	0.94	0.96	1	0.93	0.94
≥40	100%	100%	100%	100%	100%	100%	100%	100%	100%	1	1	1	1	1	1
**All**	**98%**	**98%**	**98%**	**98%**	**99%**	**98%**	**98%**	**98%**	**98%**	**0.97**	**0.96**	**0.96**	**0.96**	**0.95**	**0.95**

AA: acute appendicitis.

**Table 2 jcm-13-07257-t002:** Subset analysis of all reports for pregnant women: report prediction versus clinical conclusion.

Gestational Age (Weeks)	Total Reports	Outcome AA	Outcome Not AA	Prevalence AA	Prevalence Threshold
0–13	59	9	49	15%	13%
14–26	87	8	79	10%	0%
>26	25	4	21	16%	0%
**All**	**171**	**21**	**150**	**12%**	**8%**
	**Report** **Suggested** **Appendicitis**	**True Positive (TP)**	**False Positive (FP)**	**Positive Predictive Value (Precision)**	**Negative** **Predictive Value**
0–13	10	9	1	90%	100%
14–26	9	9	0	100%	100%
>26	4	4	0	100%	100%
**All**	**23**	**22**	**1**	**95%**	**100%**
	**Reported** **Not** **Appendicitis**	**False Negative (FN)**	**True Negative (TN)**	**Positive Likelihood Ratio**	**Negative Likelihood Ratio**
0–13	49	0	49	50	0
14–26	78	0	78	/0!	0
>26	21	0	21	/0!	0
**All**	**148**	**0**	**148**	**149**	**0**
	**True Positive Rate** **(Sensitivity)**	**True Negative Rate** **(Specificity)**	**Accuracy**	**F_1_ Score**	**Phi Coefficient (φ)**
0–13	100%	98%	98%	0.94	0.93
14–26	100%	100%	100%	1	1
>26	100%	100%	100%	1	1
**All**	**100%**	**99%**	**99%**	**0.97**	**0.97**

AA: acute appendicitis.

**Table 3 jcm-13-07257-t003:** Reported appendix diameter from each MRI report compared to final clinical outcome of either AA or not AA.

Measured Appendix Diameter (mm)	Total Count	Outcome AA (%)	Outcome Not AA (%)
<5	583	0/583 (0%)	583/583 (100%)
5	1576	1/1576 (0.1%)	1575/1576 (99.9%)
6	650	20/650 (3%)	630/650 (97%)
7	179	82/179 (46%)	97/179 (54%)
8	99	82/99 (83%)	17/99 (17%)
9	83	78/83 (94%)	5/83 (6%)
10	107	104/107 (97%)	3/107 (3%)
>10	188	188/188 (100%)	0/188 (100%)
Not measured	13	2/13 (15%)	11/13 (85%)
Total	3478	557/3478 (16%)	2916/3478 (84%)

AA: acute appendicitis.

## Data Availability

The de-identified data analyzed by us are not publicly available; however, requests for access to the data can be directed to the corresponding author and will be evaluated on a case-by-case basis.

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
