# Peer review of "Open-Access 12-Minute MRI Screening for Acute Appendicitis: A Five-Year Retrospective Observational Study of Diagnostic Accuracy"

_jcm, 2024, doi:10.3390/jcm13237257_

Round 1
Reviewer 1 Report
Comments and Suggestions for Authors
Dear authors, I really liked your work and approach to statistical feedback. It is rare to find such thoroughness. For my part, I would probably suggest reducing the number of tables (or converting them to a graphical visualization format), since there is too much data for a journal article, in my opinion. The only regret is that the use of this technology in most countries will be too expensive for its introduction into general practice. Otherwise, I have no questions.
Reviewer 2 Report
Comments and Suggestions for Authors
Overall Recommendation:
The article presents a compelling case for MRI’s efficacy as a first-line screening tool for appendicitis, contributing valuable data on diagnostic accuracy. Minor revisions are recommended to enhance clarity, address single-radiologist limitations, and add more context around comparison metrics.
Title and Abstract
The title accurately reflects the study’s aim, target population, and main investigative focus.
Abstract: it could benefit from a more explicit mention of statistical significance to provide a well-rounded summary.
Introduction
Expanding on the limitations of traditional imaging options (e.g., CT and ultrasound) would enhance the rationale for developing an MRI-based protocol as a first-line imaging option. The addition of relevant recent studies or statistics supporting MRI use could strengthen the background.
Methods
I suggest including justification for the choice of specific sequences or imaging parameters could enhance comprehension for readers.
Results
The results are well-organized, with clear reporting of sensitivity, specificity, and predictive values across different patient subgroups.
Figures and tables effectively support the results and are especially helpful in illustrating report finalization times and accuracy metrics.
Discussion
Further limitations, such as the lack of a randomized design and potential reviewer bias, could be discussed more explicitly. : The study would benefit from a dedicated section on recommendations for future research, specifically addressing the generalizability of the MRI protocol to other sites or radiologists.
Author Response
Please see that attachment

Reviewer 3 Report
Comments and Suggestions for Authors
This diagnostic accuracy study is highly relevant due to the large number of patients included (which mitigates the retrospective nature of the study) and the calculation of standardized accuracy parameters.
The methodology is described in detail and supports the reproducibility of the study, as does the description of the performance measures. The results are well-presented and thorough.
Reviewer 4 Report
Comments and Suggestions for Authors
In the manuscript, the authors present a retrospective study regarding the accuracy of abdominal MRI in the diagnoses of appendicitis. The authors proposed a 12 minute MRI protocol for the diagnoses of the acute appendicitis in the emergency department. They included in the study 3578 reports. In my opinion, it is an interesting manuscript. In order to improve the quality of the manuscript, some changes have to be done. My observation are :
- it is not clear in the manuscript what clinical form of appendicitis, the authors are referring to ? Probably is about acute appendicitis.
- maybe, the paragraph in which the authors present the statistical performance measures should e excluded from the manuscript. The formulas that are presented here are well known in the literature.
- the authors presented 8 cases of false negative results. It is not clear in the manuscript what protocol was used in the cases with negative results on the MRI imaging. This patients probably were discharge. In cases of false negative results, how was the diagnosis of appendicitis establish ?
- it is not clear in the manuscript, if the authors recorded cases of chronic appendicitis. Do you have cases with the diagnoses of appendicitis that were operated and the hystopathological result of the specimen to be chronic appendicitis ?
- all the patients with suspicion of acute appendicitis on MRI imaging were operated ?
Author Response
Please see that attachment

Round 2
Reviewer 4 Report
Comments and Suggestions for Authors
In the manuscript, the authors present a study regarding the utility of abdominal MRI in the diagnoses of acute appendicitis in the emergency department. The manuscript has been reviewed before and the authors changed the manuscript according to the previous reviewers indications. Their comments are quite pertinent. The quality of the manuscript hasd been improved. That is why, I think that this manuscript can be published in this form.